# Existence of log-phase *Escherichia coli* persisters and lasting memory of a starvation pulse

Mikkel Skjoldan Svenningsen[1], Sine Lo Svenningsen[2], Michael Askvad Sørensen[2] , Namiko Mitarai[1]

**The vast majority of a bacterial population is killed when treated with a lethal concentration of antibiotics. The time scale of this killing is often comparable with the bacterial generation time before the addition of antibiotics. Yet, a small subpopulation typically survives for an extended period. However, the long-term killing dynamics of bacterial cells has not been fully quantified even in well-controlled laboratory conditions. We constructed a week-long killing assay and followed the survival fraction of *Escherichia coli* K12 exposed to a high concentration of cipro-floxacin. We found that long-term survivors were formed during exponential growth, with some cells surviving at least 7 d. The long-term dynamics contained at least three time scales, which greatly enhances predictions of the population survival time compared with the biphasic extrapolation from the short-term behavior. Furthermore, we observed a long memory effect of a brief starvation pulse, which was dependent on the (p)ppGpp synthase *relA*. Specifically, 1 h of carbon starvation before anti-biotics exposure increased the surviving fraction by nearly 100-fold even after 4 d of ciprofloxacin treatment.**

## Introduction

Bacterial populations are quickly decimated during a typical an-tibiotics assault. Within a few generation times, the far majority of cells will be dead. However, it is typically recommended to use extended durations of treatment, ranging from several days to months, prolonging the exposure time of bacterial pathogens to the antibiotic (1). The WHO is now considering the benefits of short-ening the duration of antibiotics administration while still keeping the treatment effective because of concerns of increasing antibiotic resistance occurring as a consequence of increased exposure (1). To find the optimal treatment duration, one needs to understand the killing dynamics of bacteria when exposed to antibiotics, especially the bacterial cells surviving for longer times. The long-term sur-vivors are typically referred to as persister cells, a subgroup of cells that survive antibiotics for an extended period compared to the average of the population but have not acquired mutations that make them resistant to the antibiotic (2, 3). Most research on persister cells is performed within the well-defined conditions of the laboratory, but despite these strongly simplified conditions, and more than 70 yr of research, laboratory persisters are still far from understood (4, 5).

One pending question is whether and how much persisters form spontaneously during the exponential growth phase. Such per-sisters are called type-II (3) or spontaneous (4) persisters. It was repeatedly shown that stress-triggered (or type I) persisters are formed in high numbers during the stationary phase, but research on spontaneous persister formation during the exponential phase is sparse (4). The research has mostly been confounded by a lack of careful attention to the presence of stationary-phase cells carried over from the starter cultures, which artificially elevated the per-sister fraction of exponential cultures (4, 5, 6). One carefully exe-cuted study showed that no *Escherichia coli* persister cells were formed during fast exponential growth in rich medium (7), whereas other studies merely showed reduced levels during exponential growth (3, 5). A benefit of analysing the exponential growth phase is the well-defined physiology of this state (8, 9). This makes it possible to vary the growth physiology in a controlled manner, especially by varying the growth rate through culturing bacteria in media of different nutrient quality. It was previously shown that the bacterial growth rate strongly correlates with the death rate during the initial period of killing with $\beta$-lactams (10, 11, 12). This poses the additional questions of how the growth rate at the time of anti-biotics exposure affects the short- and long-term killing dynamics.

The current standard for persister identification at the pop-ulation level is that the killing curve is at least biphasic, where two time scales are identified in the time–kill curves (4). Persisters are identified as the subpopulation with a second, slower killing rate than the rapid death rate of the primary population. If only two time scales are present in the killing dynamics, the population survival time can be extrapolated from the second slow killing rate. Notably, the presence of more than two phases has been demonstrated previously in a few studies (3, 13, 14, 15). These observations mo-tivate the importance of studying the long-term survival of the antibiotics-tolerant subpopulation, which may not agree with ex-trapolation from short-term survival. However, most in vitro

[1]The Niels Bohr Institute, University of Copenhagen, Copenhagen, Denmark   [2]The Department of Biology, University of Copenhagen, Copenhagen, Denmark

Correspondence: mas@bio.ku.dk; mitarai@nbi.ku.dk

laboratory research on persisters of fast-growing bacteria as *E. coli* is carried out for 3–5 h (5, 7, 16), although some studies increase the exposure time to 24–50 h (2, 3, 15, 17). Investigation of long-term survival beyond the typical 5-h persister assay might reveal new insights into bacterial killing dynamics that are relevant for the week-long antibiotics treatment of bacterial infections recommended by the WHO (1).

Last, the molecular mechanism(s) of persister formation is still unknown (12, 18). Many intracellular components have been proposed to play a role (16, 19, 20, 21, 22, 23), but so far no single mechanism convincingly explains persister formation. In fact, bacterial persistence presents as a very complex and diverse problem, where the survival fraction could be composed of different subpopulations.

Despite the complexity of persistence, it has been established that stationary-phase cultures contain a greater persister fraction than exponentially growing cultures (5). Stationary-phase bacteria may refer to bacteria in a multitude of different physiological states but is typically associated with starvation stress (24). Furthermore, the second messenger (p)ppGpp, which accumulates during starvation responses, was frequently shown to correlate positively with persistence formation (5, 17, 21, 25, 26). Hence, it is critical for persistence research to understand the degree to which (p)ppGpp levels affect persistence, and under which circumstances.

The present study investigates persistence in the balanced exponential growth phase where (p)ppGpp levels are relatively low and correlate inversely with the growth rate. It deals with whether *E. coli* forms spontaneous persisters in the exponential phase, their dependence on the growth rate, how long they survive and how their formation relates to (p)ppGpp levels. We followed the long-term survival of *E. coli* K12 populations exposed to a lethal concentration of ciprofloxacin for 1 wk. The growth rate of the *E. coli* population at the time of antibiotics exposure was varied using growth medium with either of two different carbon sources. In addition, a knockout strain was constructed in the wild-type background, removing the gene *relA* and, thus, introducing a (p)ppGpp synthesis deficiency. Furthermore, we compared the killing dynamics with and without a short carbon starvation period immediately before the killing assay. The starvation pulse had a considerable influence on persister formation. This triggered persistence had a very long memory effect on the survival of the population.

## Results

### Long-term persister assay of exponentially growing cells

First, we investigated whether long-term persister cells form during exponential growth in glucose minimal medium. We treated balanced cultures of *E. coli* K-12 with ciprofloxacin and monitored the killing dynamics for 1 wk of antibiotics treatment. Balanced growth was obtained by culturing the cells for more than 20 doubling times in the target medium at 37°C, keeping the cell density of the culture below an OD$_{436}$ of 0.3 by repeated back dilutions. In total, at least $10^9$-fold dilutions were performed during the exponential phase to

ensure that the reported persister cell numbers are well above the possible number of carry-over cells from stationary phase. Cultures were then treated for a week with 10 µg/ml ciprofloxacin and their killing dynamics were monitored by repeated platings of culture aliquots on antibiotics-free growth medium (See the Materials and Methods section for details).

*E. coli* persisters were formed during exponential balanced growth in glucose minimal medium, as seen in Fig 1. There was a fast initial killing at a rate of about 1/0.3 (h$^{-1}$), with a slower killing rate already after 2 h. When we fit a biphasic curve (summation of two exponential functions) to the data up to 7 h, the second phase of killing is at a rate of about 1/2.3 (h$^{-1}$), shown by a brown line in the inset in Fig 1. However, for longer times, this fit significantly underestimates the survival time of the bacterial population (Fig 1). In other words, there is an even slower phase of killing, extending from 7 h to 4 d. Last, from day 5 to day 7, the remaining cells were killed at a very slow rate; however, this part of the data is less reliable because of the small numbers of recovered colonies.

To identify the various phases in a quantitative manner, the data were fitted with a sum of exponentials, and the appropriate number of exponentials was chosen with a $\chi^2$-test (see the Materials and Methods section). The double exponential, which corresponds to the biphasic killing dynamics, was rejected as a good fit by the test. Instead, a fit using the triple exponential functions was accepted. The best fit obtained was the first phase of killing at a rate 1/0.3 (h$^{-1}$) (a cyan dashed line), the second phase of killing at a rate of 1/8 (h$^{-1}$) (a green dashed line), and the third phase of killing at a rate of 1/57 (h$^{-1}$) (a blue dashed line). The long-term killing dynamics thus had more than two time scales, which is only apparent after several days of measurement.

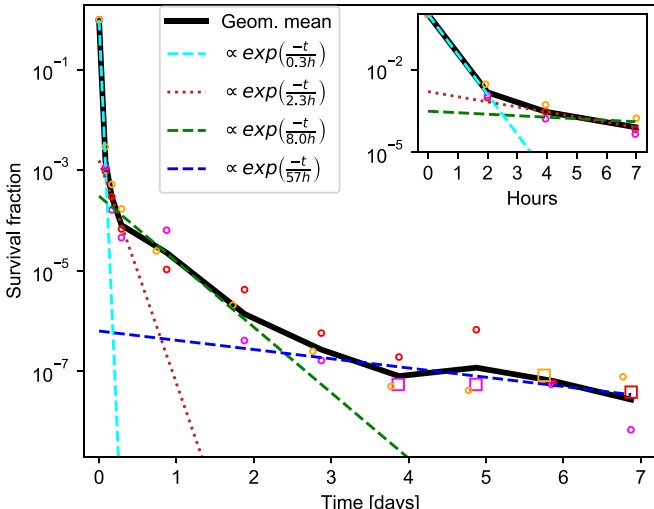

**Figure 1. Killing dynamics of exponential-phase *Escherichia coli* persisters.**
Killing dynamics for exponential-phase persisters. The bacteria were grown in glucose minimal medium. All three biological replicates are shown for each data point (circle). When zero colonies are recovered, the detection limit is shown (square, see the Materials and Methods section for details). The black line is the geometrical mean. Each separate phase of a triphasic fit is shown, respectively, as cyan, green, and blue dashed lines. The second phase of a biphasic fit to the first 7 h is shown as a brown dotted line.

The exponential-phase growth rate determines many aspects of bacterial physiology, including the macromolecular composition ([8], [27]). The growth rate at the time of antibiotics exposure has previously been linked to short-term survival of antibiotics ([10], [11]) and could also affect long-term survival. For that reason, the long-term killing assay was repeated with glycerol as the carbon source, which strongly affected the wild-type growth rate. In glucose minimal medium, the wild-type doubling time was $50 \pm 1.4$ min, whereas it was $106 \pm 3.0$ min in glycerol minimal medium. This difference had an impact on the initial phase of killing for up to 7 h,

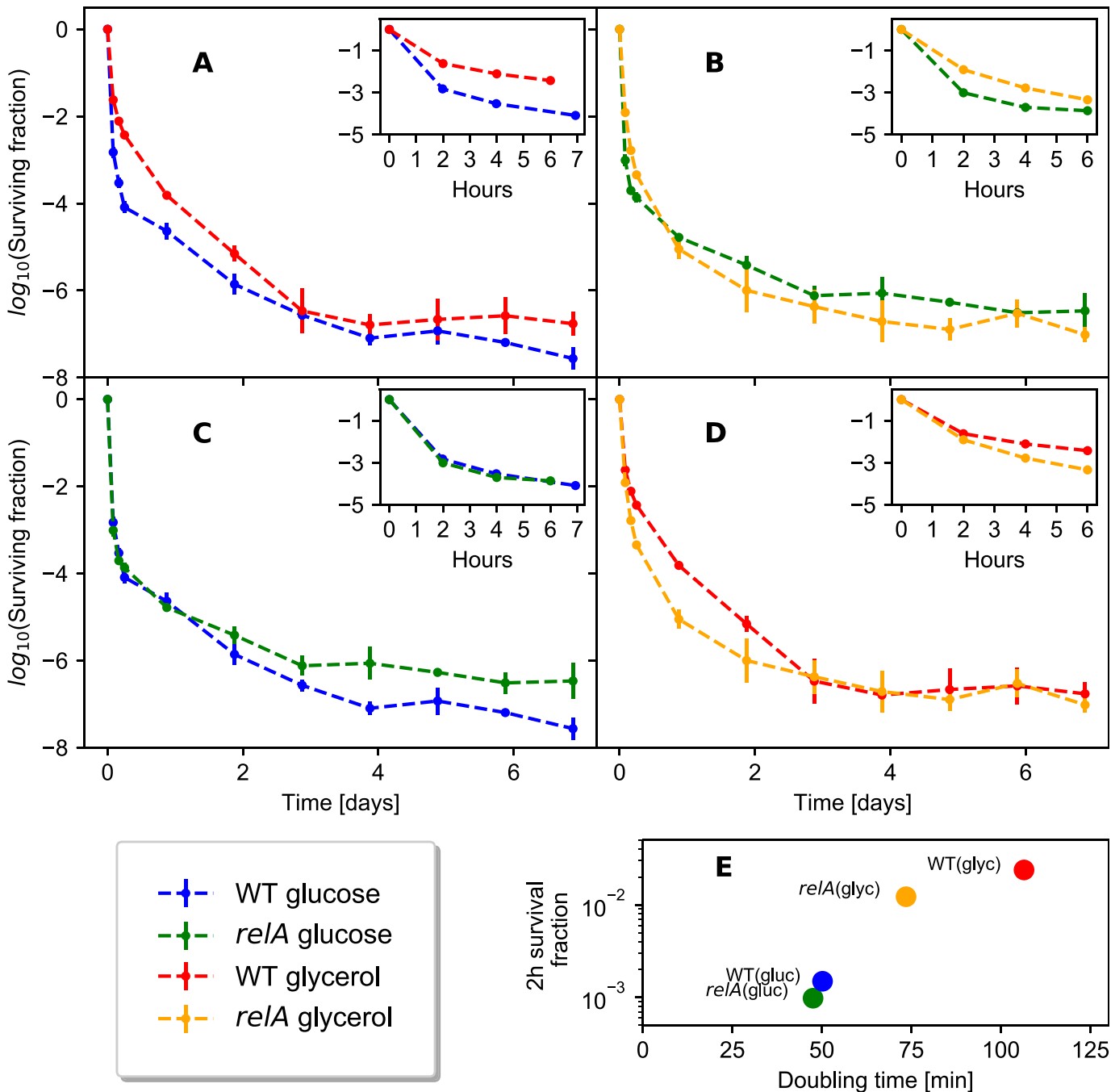

**Figure 2. Killing dynamics in different media with and without the *relA* gene.**
Each line represents a geometric average of three biological replicates. **(A)** The wildtype killing dynamics in either glycerol or glucose minimal medium. **(B)** The ΔrelA killing dynamics in either glycerol or glucose minimal medium. **(C)** Killing dynamics in glucose minimal medium for comparison between the wildtype and the ΔrelA strain. There was no statistically significant difference between the wildtype survivors and the ΔrelA strain survivors at any of the timepoints (see the Materials and Methods section). **(D)** Killing dynamics in glycerol minimal medium for comparison between the wildtype and the ΔrelA strain. **(A, B, C, D)** Log–log plot of the data in (A, B, C, D) is available in Fig S1. **(E)** Survival fraction after 2 h of ciprofloxacin treatment plotted against the growth rate before antibiotics treatment.

showing slower killing and a higher survivor fraction in glycerol (Fig 2A inset).

However, the long-term killing curve in glycerol minimal medium was not merely a decelerated version of the killing curve in glucose minimal medium. Although significantly more cells survive in glycerol than in glucose for up to a day (Fig 2A), the survival curve in glycerol medium from 7 to 21 h had a steeper slope than in glucose medium, resulting in comparable surviving fractions after 2–3 d. In fact, in two of the three biological replicates of wild-type cultures grown in glycerol, almost no survivors were observed after 3 d of killing (see Fig S2), supporting further that the survival in glycerol minimal medium is not more than that in glucose minimal medium after 3 d.

Overall, the wild-type killing dynamics in glycerol had more than two time scales, with a best fit of four separate phases of killing (Fig S3).

### Deletion of *relA* affects the killing dynamics

In glycerol minimal medium, the steady state (p)ppGpp level of the wild-type strain is higher than in glucose minimal medium (28). Because the (p)ppGpp level has been frequently associated with persister formation, we next aimed at investigating the effect of the initial (p)ppGpp level on long-term survival. *E. coli* encodes the primary (p)ppGpp synthetase RelA and the secondary (p)ppGpp synthetase SpoT, the latter of which is bifunctional as a (p)ppGpp hydrolase. The nutrient-dependent steady-state growth rates are inversely related to the concentrations of (p)ppGpp, both for *relA*[*] and *relA*[−] strains (28), and for many carbon sources, the growth rates and (p)ppGpp levels of *relA*[*]/*relA*[−] strain pairs are indistinguishable because of (p)ppGpp synthesis by SpoT. However, in low-energy carbon sources such as glycerol or acetate, SpoT produces insufficient (p)ppGpp to reduce the growth rate when RelA is missing, leading to an enhanced growth rate of the RelA mutant strain relative to the wild type (28). We constructed a Δ*relA* mutant to clarify the role of (p)ppGpp in the killing dynamics.

As expected, the difference between the growth rate in glucose and in glycerol minimal medium was smaller for the Δ*relA* strain than the wild type, with a doubling time of 47 ± 1.5 min in glucose minimal medium and 74 ± 1.6 min in glycerol minimal medium. The survival after 2 h is positively correlated with the doubling time (Fig 2E), consistent with the previous observations that the initial killing rate decreases with the doubling time (11). The short-term (up to 4 h) survival under antibiotics exposure was also correlated with the growth rate in the Δ*relA* strain (Fig 2 inset).

Like wild-type cells, the Δ*relA* mutant formed long-term survivors in both glucose and glycerol minimal medium with more than two phases of killing (Fig 2B, see the Materials and Methods section for statistical analysis). In glucose, the two strains grew at similar rates, and the early killing dynamics of the Δ*relA* mutant were very similar to that of the wild type (Fig 2C). In glycerol, the faster growing Δ*relA* mutant showed a significantly lower level of persisters than the wild type in the initial phase of killing up to 1 d (Figs 2D and S1), indicating the importance of growth rate, or ppGpp level, for persister formation in this phase. Interestingly, the long-term survival of the Δ*relA* mutant and wild type were comparable at later times (after 3 d) in glycerol medium, and somewhat higher

than the wild-type strain in glucose medium. Thus, survival in the long term is not simply dependent on the population growth rate at the time of antibiotics exposure.

### A starvation pulse before the antibiotic application affects the long-term persistence of wild-type cells

A sudden downshift of the carbon source is known to give a spike of the (p)ppGpp level in the wild-type strain just after the downshift, whereas the spike is significantly lower in a RelA strain (29).

We then wondered if a short pulse of carbon source starvation to the exponentially growing cells before the killing assay would give a quantifiable difference in the long-term persistence between the wild-type strain and the Δ*relA* strain. If the starvation pulse increases the persisters, it would be considered as triggered persistence, and the current study would allow us to quantify how long such triggered persisters last.

To test this, part of the cultures in balanced growth were filtered into growth medium without a carbon source and starved for 1 h, before the carbon source was replenished (Fig 3A). Fig 3B shows that the 1-h starvation pulse resulted in a quick rise of the ppGpp level peaking at about 15 min after the downshift for the wildtype strain in glucose medium, whereas only a mild increase in the ppGpp level was seen in the Δ*relA* strain. Antibiotics were added simultaneously with the carbon source replenishment (Fig 3A, see the Materials and Methods section for details). Remarkably, the short carbon starvation before the addition of antibiotics had long-term effects on the killing dynamics. This was especially visible for the wild-type strain grown in glucose minimal medium, where the brief starvation period reproducibly resulted in almost 100-fold more persisters for up to 4 d (Fig 4A). The difference was abolished by removing *relA*, as seen in Fig 4B; the Δ*relA* strain only exhibited increased survival for the first 6 h after starvation. As such, the long-term memory of the starvation pulse is seemingly a *relA*-dependent effect. However, the Δ*relA* strain in the steady-state growth in glucose had more long-term survivors than the wild-type strain, and the downshift brought the wild-type strain survival fraction to a level similar to the Δ*relA* strain.

The effect of the downshift was smaller in the glycerol medium (Fig 4C and D). In the wild-type strain, the average persister level with downshift was higher up to 4 d, but the statistical significance of the difference was confirmed only up to 7 h because of the larger data scatter for later time points (Fig 4C). The effect of downshift disappeared faster in the Δ*relA* strain already after 2 d (Fig 4D).

## Discussion

We expanded the understanding of bacterial killing dynamics with a long-term persister assay. The use of minimal medium facilitated the formation of long-term persisters during exponential growth, in contrast to growth and killing in rich medium (7). Spontaneous persisters were observed during the exponential growth phase, both in glucose and glycerol minimal medium, and in some cases, they survived at least 1 wk. This long-term survival does not require *relA*, although the residual (p)ppGpp synthesized by SpoT is likely

▶▶▶ Life Science Alliance

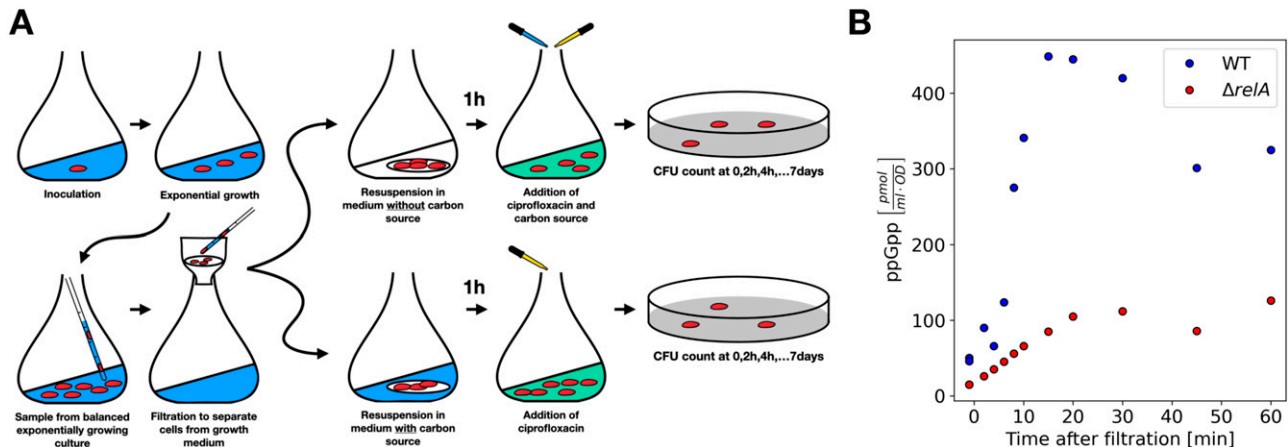

**Figure 3.  The experimental setup for the long-term persistence assay.**
**(A)** Cells were in balanced exponential growth before the killing assay. A part of the culture was filtered and then resuspended in medium with or without the carbon source, and incubated for 1 h. After 1 h, the ciprofloxacin was added to both of the samples, and at the same time the carbon source was added to the culture that has been starved for 1 h. The first sample was taken just before the additions. After that, samples were taken at 2, 4, 6/7, 21 + 24 n hours after the antibiotic addition for $n \in [0;6]$. The samples were washed and plated on agar plates containing the target medium. **(B)** The 1-h time course of ppGpp level for the culture grown in the glucose medium, filtered, and resuspended in fresh medium without carbon source. The time zero is the time of the resuspension. The wild type (blue circles) shows clear peak around 15 min after the resuspension. ΔrelA strain shows only mild increase in the ppGpp level. The phosphoImager scan of TLC plates used to quantify ppGpp levels is shown in Fig S4

necessary. In fact, there is an increase in long-term survival of the ΔrelA mutant in glucose.

We have shown that a 1-h starvation pulse before the addition of the antibiotic affects long-term survival. The finding that a short starvation pulse gives a long-term effect is consistent with a previous study of a temporal nitrogen downshift before antibiotics treatment, which was shown to elevate the persister level at 24 h in a relA-dependent manner (30). Also, a few other studies have previously shown that triggered persistence can be relA dependent (21, 25). Our study demonstrated that the memory can be remarkably long-lasting, as 1-h carbon starvation gave an increase in survival for at least 4 d in the wild-type strain grown in glucose medium. The molecular mechanism underlying this long-term memory is yet to be investigated, but in all likelihood it is linked to the abrupt RelA-mediated rise in (p)ppGpp upon starvation because the starvation-pulse effect on long-term survival was abolished in the ΔrelA mutant. In further support of this hypothesis, there was no long-term effect when glycerol was used as the carbon source, which could be due to the high basal level of (p)ppGpp in glycerol relative to glucose minimal medium (8). The sensitivity of the survival fraction to the rather short perturbation may be consistent with the idea that there is a threshold in some molecule concentration to determine if the cell becomes a persister or not (31, 32) because a small perturbation can have a major impact on the occurrence of rare expression patterns that exceed an extreme threshold (33). This observation also alerts us that a small perturbation in the experimental procedure may strongly affect the result of persister assays.

This study shows that the long-term killing of E. coli in ciprofloxacin is not adequately described by biphasic dynamics. At least three phases of killing were present in the data. Thus, despite the emphasis on at least a biphasic behavior to define persistence (4), a third, or even fourth, phase of killing occurs that may even be more

clinically relevant. The presence of additional phases also means that the population survival time will be underestimated by predictions from the biphasic killing assumption. Indeed, it is not sufficient to measure killing dynamics for only 5 h and then extrapolate the population survival time from there.

The detailed molecular mechanism of the observed persistence is not the focus of the current study. Nevertheless, it is worth mentioning that ciprofloxacin has been reported to induce persistence via toxin activation through the SOS response in the killing dynamics up to 6 h (34). The existence of more than two killing phases indicates that different mechanisms may play roles for longer term persistence on top of the previously studied ones. For the future study of the molecular mechanisms of persistence, attention should be paid to which time scale of the survival the pathway affects.

The population growth rate was found to be positively correlated with the killing rate in the initial phase. However, the correlation was diminished in the longer term survival, and lost in the third phase of killing. In shorter persister assays, a difference in growth rate, such as between different mutants, might strongly confound results when analysing persistence fractions. These differences seem smaller and less relevant in later phases of killing.

This investigation of long-term killing dynamics has added to the concept of bacterial persistence as a complex phenomenon. During long-term antibiotics treatment, different mechanisms could account for bacterial survival on different time scales (an hour, a day, a week), although the (p)ppGpp level at the time of initial exposure to the antibiotic seems to be important in all cases. As such, persistence seems to be a time-dependent phenomenon, where different survival mechanisms account for different bacterial life spans. The presence of several phases in the killing dynamics begs the question if a more extended concept should replace the simplified concept of bacterial persistence.

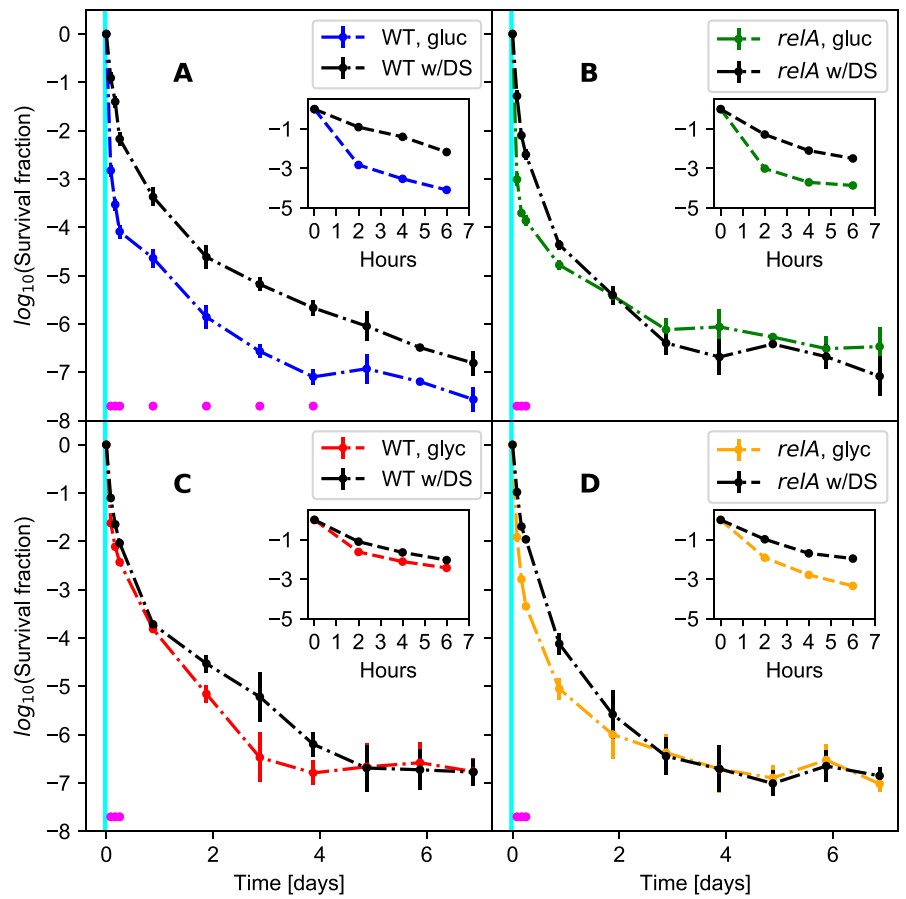

**Figure 4. The effect of a short starvation-pulse on the survival dynamics.**
All strains and conditions are shown with and without the downshift. The black line is with the downshift, the colored line is without the downshift. The magenta dots represent statistically different data points as tested by an unequal variances two-sided $t$ test. The cyan interval illustrates the downshift period before the killing assay. **(A)** The wild type in glucose. A downshift in glucose significantly enhances wild-type survival for 4 d and seems to be increased for up to all 7 d. **(B)** The $\Delta relA$ mutant in glucose. **(C)** The wild type in glycerol. **(D)** The $\Delta relA$ mutant in glycerol.

## Materials and Methods

### Strains

The strain MAS1081 (MG1655 $rph^+gatC^+glpR^+$) was used as the wild type (35). The $\Delta relA$, MAS1191, is MAS1081 made $\Delta relA251::Kan$ by P1 transduction from CF1651 (36) followed by selection on kanamycin.

### Long-term killing assay

A single colony of the *E. coli* strain grown on an agar plate was incubated overnight in the target medium (MOPS minimal medium with either glucose or glycerol as the carbon source (37). See the next subsection for the recipe). This was performed for each biological replicate. The overnight culture was diluted 1:10$^7$ in 10 ml target medium in a 100-ml Erlenmeyer flask. The flask was continuously shaken at 160 RPM by a shaker platform (New Brunswick Innova 2300 [orbit 5.1 cm], Eppendorf, Germany) in a 37°C room. Hours later, the culture was diluted further, at least 1:10$^2$, in 100 ml preheated medium in a 1-L Erlenmeyer flask, reaching a total dilution of at least 1:10$^9$. The $OD_{436}$ of the cultures never exceeded 0.3 because of back dilutions to ensure the balanced growth. The growth rates were calculated based on at least five measurements

in the measurable $OD_{436}$ interval from 0.03 to 0.3 (Fig S5 and Table S1). Carbon starvation was introduced by filtration of the culture (typically around 20 ml), followed by immediate resuspension of the cells on the filter in 40 ml of the preheated target medium (with or without carbon source) in a 300-ml Erlenmeyer flask. As such, both the control and the starved culture were exposed to filtration and resuspension. The medium volume never exceeded 14 percent of the flask volume. The starvation was verified by measuring $OD_{436}$ twice during the 1-h downshift, to confirm either an increase (control culture) or no change (starving culture) in biomass. The $OD_{436}$ measurements verified that a starvation took place for all the experiments that were supposed to experience a downshift. It also verified that growth, with a doubling time approximately close to the expected value, took place in the control cultures.

After 1 h of starvation, a sample was taken immediately before ciprofloxacin (10 µg/ml) was added along with the carbon source, which was replenished to end the downshift. Most of the samples were taken at times 2, 4, 6/7, and 21 + 24·n hours for n ∈ [0;6] (see next subsection for the detail of the sampling timing). The samples were put on ice for a few minutes and then centrifuged for 10 min at 4°C at 10,000$g$. The supernatant was removed, and the cell pellet was resuspended in room temperature MOPS buffer with no supplements. The sample was diluted appropriately, never more than 1:10$^2$ per step, corresponding to 10 in 990 µl. The sample was

plated with 200 μl per plate on minimal medium plates containing the target medium. The plates were kept at 37°C for at least 1 wk and all colonies were counted. The whole experiment is illustrated in Fig 3. Detection limit of each experiment is presented in Fig S6.

### Correction for difference in sampling times

Most samples were obtained at approximately hours 2, 4, 6, 21 + 24·n hours for n ∈ [0;6]. The wild-type glucose experiments were sampled at approximately 2, 4, 7, 21 + 24·n, but one of the experiments in glucose had the following time points 2, 4, 7, 18 + 24·n where n ∈ [0;6]. We corrected this dataset, by calculating a straight line between the data point of interest and the following data point in log space. We used this straight line to extrapolate the value to the timepoint of interest. The original data values and corrected values are presented in Table S2. All statistical analyses were carried out with and without this correction, to make sure it did not affect the results.

### Growth medium

According to reference 37, MOPS minimal medium was made by mixing the following:

(i) 100 ml 10× MOPS medium that contains 40 mM MOPS, 4 mM tricine, 1.32 mM $K_2HPO_4$, 9.52 mM $NH_4Cl$, 0.523 mM $MgCl_2$, 0.276 mM $K_2SO_4$, 0.01 mM $FeSO_4$, 0.5 μM $CaCl_2$, 50 mM NaCl, and trace metals (see reference 37).
(ii) Carbon source (one of either)

  (1) 10 ml 20% glucose (for 0.2% wt/vol glucose minimal medium)
  (2) 8 ml 50% glycerol (for 0.4% vol/vol glycerol minimal medium). The 50% glycerol stock solution is stored at –20°C.
(iii) Fill up to 1 L by adding milliQ water

Mops minimal medium plates were made by adding 15 g Bacto agar (autoclaved with the milliQ water) to the Mops minimal medium recipe above.

### Controls

After each completed experiment (7 d), 200 μl of culture, still containing antibiotics, was spread on a plate with the target medium. This was left at 37°C for at least 7 d, and no growth confirmed the absence of a growing resistant culture in the flask. In addition, the activity of the antibiotics in the culture was tested after 7 d by dropping 20 μl on a lawn of growing *E. coli*.

### Data from each biological replicate

Each biological replicate of the glucose experiments are shown in Fig S7, whereas the replicates of glycerol experiments are shown in Fig S2. If the observed CFU was zero (no colonies counted), the value zero was replaced by the detection limit defined as one colony per plate.

### Analysis of time scales in the killing dynamics

Time scales in the killing dynamics were statistically identified by fitting a sum of exponential functions to the data. The model with the least number of parameters, that could not significantly be rejected, was then chosen (38). The functional form of the models was a sum of exponential functions. The number of exponential functions in the sum corresponded to the number of time scales. A biphasic killing curve would, for example, be well fitted by the sum of two exponentials. The $\chi^2$-test for goodness of fit is used for identifying the appropriate model using the Minuit minimization software (39, 40). The functional form of the models are as follows:

$$Model1(t, a_0) = \exp(-a_0 t),$$

$$Model2(t, a_0, a_1, b_1) = (1 - b_1)\exp(-a_0 t) + b_1\exp(-a_1 t),$$

$$Model3(t, a_0, a_1, b_1, a_2, b_2) = (1 - b_1 - b_2)\exp(-a_0 t) + b_1\exp(-a_1 t) + b_2\exp(-a_2 t),$$

$$Model4(t, a_0, a_1, b_1, a_2, b_2, a_3, b_3) = (1 - b_1 - b_2 - b_3)\exp(-a_0 t) + b_1\exp(-a_1 t) + b_2\exp(-a_2 t) + b_3\exp(-a_3 t).$$

Each exponential function contains a specific time scale which corresponds to each exponent. A biphasic killing curve would, for example, be well fitted by the sum of two exponentials. To statistically determine which model fits to the data, each model is fitted to the data by using least squares and the Minuit optimization software (39, 40). The $\chi^2$ is then calculated for each of these models. The degrees of freedom correspond to the number of data points with the subtraction of the number of parameters for fitting and one degree of freedom for normalizing the data. Thus, the probability that the data do not correspond to the model is calculated using the $\chi^2$ cumulative distribution. The *P*-value was chosen to be 0.05. For the wild-type strain grown in glycerol, the Model 4 was an appropriate fit, for the three other datasets, the Model 3 was an appropriate fit. The parameters estimated are given in supplement in Table S3. The fits are shown in the supplement in Fig S3.

### Display of fits in Fig 1

The functions shown in Fig 1 all stem from the fit of Model 3 to the data. Model 3 is given by the following equation:

$$Model3 = (1 - b_1 - b_2)\exp(-a_0 t) + b_1\exp(-a_1 t) + b_2\exp(-a_2 t).$$

The functions displayed are then given by the three terms of the fit shown separately, as given by:

$$term1 = (1 - b_1 - b_2)\exp(-a_0 t),$$

$$term2 = b_1\exp(-a_1 t),$$

$$term3 = b_2\exp(-a_2 t).$$

This is to illustrate the contribution from each exponential time scale.

In addition, the second phase of a biphasic fit to the data shown in the inset, is also shown in the main figure and in the inset.

## Statistical analysis

All killing curves are based on three biological replicates. The mean is determined as the mean of the logarithmic values, which corresponds to the geometric mean. This is done to get a more adequate mean-value representation in log space. The uncertainties are also calculated as the standard deviations of the log-transformed values. Whenever a data point had the value zero, which happened frequently, that value was replaced with the detection limit, to get a sensible value in log space.

An unequal variance two-sided $t$ test was used to determine significant differences between two data points at the same time point ($P < 0.05$).

## ppGpp measurements

The measurements were performed essentially as described in reference 41 and used in references 42, 43, and 44. In short, cultures were grown for two generations in the presence of 75 $\mu$Ci/ml 32P-phosphate at a total phosphate concentration of 0.33 mM. At the time of starvation, cultures were filtered, washed in medium without glucose and phosphate and resuspended in the medium without glucose but containing 32P at the same specific activity as during growth. These steps were performed at 37°C and lasted less than 2 min. For determination of the nucleotide pools, 100 $\mu$l of culture was harvested into 20 $\mu$l 2 M formic acid at 0°C. After centrifugation, the nucleotides in the supernatant were separated by chromatography on polyethyleneimine-cellulose plates. The activities of the individual spots were quantified by PhosphoImager scans (Typhoon Phosphor Imager FLA7000 [GE Healthcare]) of the plates. The specific activity of the signal was determined from a medium sample from the individual cultures spotted onto the same brand of plates that were exposed together with the chromatograms.

## Supplementary Information

## Acknowledgements

MS Svenningsen and N Mitarai thank S Semsey for fruitful discussions. This work was supported by the Danish National Research Foundation (DNRF120), the Independent Research Fund Denmark (8049-00071B and 8021-00280A), and the Villum foundation (00028054).

## Author Contributions

MS Svenningsen: conceptualization, data curation, formal analysis, investigation, visualization, methodology, and writing—original draft, review, and editing.

SL Svenningsen: conceptualization, resources, supervision, funding acquisition, and writing—review and editing.

MA Sørensen: conceptualization, resources, data curation, formal analysis, supervision, funding acquisition, investigation, visualization, methodology, and writing—review and editing.

N Mitarai: conceptualization, resources, formal analysis, supervision, funding acquisition, investigation, methodology, and writing—original draft, review, and editing.

## Conflict of Interest Statement

The authors declare that they have no conflict of interest.

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
