## [Reviewer comments · Life Science Alliance]

Existence of log-phase *Escherichia coli* persists and lasting memory of a starvation pulse

Mikkel Svenningsen, Sine Svenningsen, Michael Sørensen, and Namiko Mitarai
DOI: <https://doi.org/10.26508/lsa.202101076>

Corresponding author(s): Namiko Mitarai, University of Copenhagen and Michael Sørensen, University of Copenhagen

Review Timeline:

Submission Date:	2021-03-26
Editorial Decision:	2021-06-29
Revision Received:	2021-09-28
Editorial Decision:	2021-10-18
Revision Received:	2021-10-28
Editorial Decision:	2021-10-28
Revision Received:	2021-11-04
Accepted:	2021-11-04

Transaction Report:

June 29, 2021

Re: Life Science Alliance manuscript #LSA-2021-01076-T

Namiko Mitarai
University of Copenhagen
The Niels Bohr Institute
Blegdamsvej 17
Copenhagen, Copenhagen 2100
Denmark

Dear Dr. Mitarai,

Thank you for submitting your manuscript entitled "Existence of log-phase *Escherichia coli* persists and lasting memory of a starvation pulse" to Life Science Alliance. The manuscript was assessed by expert reviewers, whose comments are appended to this letter.

As you will note from the reviewer comments below, the reviewers were intrigued by these findings, but do raise some technical issues that need to be addressed. All the concerns that the reviewers have raised need to be addressed, prior to further consideration of the manuscript at LSA.

Along with clarifications, discussion points, and further quantification requests, Reviewer 2 also has the following concerns to be addressed:

- Line 169: please add some growth curves in minimal medium glucose or glycerol (for the WT and the *recA* mutant) to enrich the paper with some addition data.
- Line 212: please add the double *relA/spoT* mutant to discuss the role of (p)ppGpp (and quantify (p)ppGpp in *relA* or *relA/spoT* mutant in their condition).
- Figure 3B - To assess the validity of their experiment, please quantify (p)ppGpp in a *relA/spoT* mutant (as a control).

Thank you for this interesting contribution to Life Science Alliance. We are looking forward to receiving your revised manuscript.

Sincerely,

- A letter addressing the reviewers' comments point by point.
- An editable version of the final text (.DOC or .DOCX) is needed for copyediting (no PDFs).

B. MANUSCRIPT ORGANIZATION AND FORMATTING:

Reviewer #1 (Comments to the Authors (Required)):

The work "Existence of log-phase *Escherichia coli* persists and lasting memory of a starvation pulse " describes survival assays of *E.coli* under ciprofloxacin treatment (CIP) during balanced-growth and after a starvation trigger. The kill curves obtained in balanced growth reveal different kill rates over time. They show that whereas the initial killing rate depends on the growth rate, the long term killing is less affected. This holds whether the reduced growth rate is due to a poorer medium or a mutation (*relA*). In contrast, they show that triggered persistence (by a short amino acid starvation pulse) does depend on *relA*. In general, the work presents carefully performed experiments that are of interest to the community of antibiotic response, as they clarify previous results, in much better controlled conditions.

One main comment is the emphasis on the non-biphasic kill curve, as if antibiotic persistence is restricted to biphasic killing. The consensus refers to the definition for one sub-population of persisters, but does not exclude more than one sub-population. Many works have reported that the kill curve of wt strains contain more than a simple biphasic and several persistent sub-populations may co-exist. For example, log-phase persisters under fluoroquinolones have been described in (ref. 16). There also the kill curve is not biphasic so this reference should be mention in that context. Also, in ref. 3, the wt kill curve was fitted with more than two exponentials.

Another comment refers to the triggered persistence assay: once the bacteria are exposed to starvation, even for a short duration, they cannot be considered as "log phase bacteria", so that these observations should be placed in the context of triggered persistence.

Additional comments:

Effect of ppGpp mutants in reducing the triggered persistence of wt *E.coli* has been shown in Koch et al. *Molecular Microbiology* 2003.

Line 283 : killing curves up to 20 days in antibiotics have been done in the past (for example for MTB) so maybe avoid calling it a "new long-term persister assay".

Please define how the different biological replicates are done. Are they started each from a separate single colony?

Fig. S1: please define w/DS in legend

MOPS: is there any addition of trace elements or B1?

Reviewer #2 (Comments to the Authors (Required)):

GENERAL COMMENT:

In their paper, Svenningsen et al. explore the existence of long-term persisters in *E. coli*. The authors performed killing assays on a long time scale (1 week). Then, they assessed the role of *RelA* and the impact of nutrient starvation in the killing dynamic they observed.

Even if the first part of the manuscript about the long-term survival of bacteria has a potential interest, the general message needs to be clarify. The small amount of data and their relevance are questionable. The authors' conclusions could be entirely restricted to lab conditions. For example, *RelA*-dependent production of (p)ppGpp is required by pathogens to invade host cells and to establish a long-lasting infection (even in absence of antibiotics). Therefore, the role of *RelA* in this timescale is somehow

questionable. In addition, the authors discussed about the log-phase persisters but they did not take in consideration the role of ciprofloxacin itself in persister formation. Finally, the last part about the role of RelA in this phenomenon is superficial and does not bring any insights on persister physiology.

In conclusion, this piece of work is still preliminary. Therefore, we strongly recommend the authors to better investigate and strength the link between (p)ppGpp and long-term persistence by adding some controls (the relA/spoT mutant for example).

ABSTRACT

"...bacterial population is killed within a time scale comparable to their generation time when treated with a lethal concentration of antibiotics". This sentence is very confusing. What does "comparable to their generation time" mean?

INTRODUCTION

-Line37 - "One carefully executed study showed that no E. coli persister cells were formed during fast exponential growth in rich medium (7)". Please avoid using too many superlatives when referring to a study written by the same authors (M.A. Sorensen in both paper).

-Title of the reference 9 is incorrect.

-Line64 - "low level contamination of the persister assay". Please clarify what "low level contamination" means.

-Line78 - Reference 12 is not related to persisters. This paper refers to "non-heritable antibiotic resistance" (cfr. title) and do not fit with the standard definitions of the field (ref. 4). Please remove.

-Line90- remove "stress".

RESULTS

Long-term persister assay of exponentially growing cells

-Line128 - "OD436" - Why not OD600?

-Line133- The authors claim that E. coli persisters were formed during exponential phase. It would be nice if the authors could described th method in a cartoon. 20 doubling times should be around 1000 minutes, which correspond to approximatively 16h. Please precise this in the text. In addition, ciprofloxacin has been shown to induce bacterial persistence in E. coli (Dorr et al., 2010). Did authors try others AB (beta-lactams)?

Line148-153- "The test rejects a double exponential as a good fit, but accepts the hypothesis of a triple exponential." Could you rephrase and clarify?

Line159- "the dominant phase of killing". What doesn "dominant" mean here? Please, rephrase.

Line169- I would recommend to add - at least - some growth curves in minimal medium glucose or glycerol (for the WT and the recA mutant) to enrich the paper with some addition data.

Line178-"the survival curve in glycerol medium at 1 to 2 days had a steeper drop than in glucose medium". This is not what I see on the figure 2A. The major difference between glucose and glycerol is between the day 0 and the day1 - under the consideration it is indeed significant. Could the authors clarify this point and perform statistical analysis to quantify the significance of the differences they observe?Line183-"In fact, in two of the three biological replicates of WT cultures growing in glycerol, almost no survivors were observed after 3 days of killing". The discrepancy the authors describes by comparing their biological repeats is confusing. Please rephrase. In addition, by looking at figure 2A, no difference between glucose and glycerol at 3 days can be observed, which is not the conclusion of the authors. Please, clarify.

Deletion of relA affects the killing dynamics

Line205- "However, in low energy, SpoT produces insufficient (p)ppGpp to suppress the growth rate when RelA is missing". What does "suppress the growth rate" mean?

Line212- the authors should add the double relA/spoT mutant to discuss the role of (p)ppGpp (and quantify (p)ppGpp in relA or relA/spoT mutant in their condition).

Line216- "(Fig. 1B inset)". Do you mean Fig. 2B inset?

Figure2C - "The relA mutant had more survivors, but there was no statistically significant difference at any of the time point". If it no statistically significant, the sole conclusion should be that relA mutant does not have more survivors. Please rephrase or clarify your point.

Figure2E - Two hours treatment does not allow any convincing conclusions regarding persistence, therefore, what is the message of this panel? If any, please remove it.

A starvation pulse prior to the antibiotic application affects the long-term persistence of wt cells in glucose minimal medium.

Title: A starvation pulse done in a glucose minimal medium seems contradictory. Please rephrase.

Line248 - It is unclear why the authors filtered their bacteria instead of washing 2 times with a minimal medium containing no carbon sources. Please explain.

Figure 3B - To assess the validity of their experiment, the authors should quantify (p)ppGpp in a relA/spoT mutant (as a control).

DISCUSSION

Line 286 - "Spontaneous persisters were observed". I would recommend the authors to discuss the effect of ciprofloxacin on persister formation.

Dear Dr. Guidi,

Thank you for the constructive review. We believe that we have addressed all the concerns and the manuscript is ready for publication. We answer the reviewers' comments in detail below, but here we summarize our response to the point made in your e-mail:

- Line169: please add some growth curves in minimal medium glucose or glycerol (for the WT and the *recA* mutant) to enrich the paper with some addition data.

We have added the growth curves in the supplementary material.

-Line 212: please add the double *relA/spoT* mutant to discuss the role of (p)ppGpp (and quantify (p)ppGpp in *relA* or *relA/spoT* mutant in their condition).

- Figure 3B - To assess the validity of their experiment, please quantify (p)ppGpp in a *relA/spoT* mutant (as a control).

As we have answered to the reviewer 2 below, the double *relA/spoT* mutant does not grow in the growth media used here. Hence, we are unable to perform the same experiments with the *relA/spoT* mutant. However, please note that we have quantified (p)ppGpp for wild type and *relA* mutant. We reference previous literature wherein we or members of our laboratory have quantified (p)ppGpp by the same method (Tian, C., et al. (2016). *Journal of bacteriology*, 198(14), 1918-1926; Sinha AK and Winther KS. (2021) *Communications biology* ;4(1):1-0).

Sincerely yours,

Mikkel S. Svenningsen, Sine L. Svenningsen, Michael Sørensen and Namiko Mitarai

Reviewer #1 (Comments to the Authors (Required)):

The work "Existence of log-phase *Escherichia coli* persists and lasting memory of a starvation pulse " describes survival assays of *E.coli* under ciprofloxacin treatment (CIP) during balanced-growth and after a starvation trigger. The kill curves obtained in balanced growth reveal different kill rates over time. They show that whereas the initial killing rate depends on the growth rate, the long term killing is less affected. This holds whether the reduced growth rate is due to a poorer medium or a mutation (*relA*). In contrast, they show that triggered persistence (by a short amino acid starvation pulse) does depend on *relA*. In general, the work presents carefully performed experiments that are of interest to the community of antibiotic response, as they clarify previous results, in much better controlled conditions.

Thank you for the overall positive evaluation of our work.

One main comment is the emphasis on the non-biphasic kill curve, as if antibiotic persistence is restricted to biphasic killing. The consensus refers to the definition for one sub-population of persisters, but does not exclude more than one sub-population. Many works have reported that the kill curve of wt strains contain more than a simple biphasic and several persistent sub-populations may co-exist. For example, log-phase persisters under fluoroquinolones have been described in (ref. 16). There also the kill curve is not biphasic so this reference should be mention in that context. Also, in ref. 3, the wt kill curve was fitted with more than two exponentials.

Thank you for bringing this up. We agree that the biphasic curve consensus should refer to the definition for one sub-population and it should not exclude more than one subpopulation. Yet, it was not explicitly discussed in for example in the recent review on the consensus on persister research (Balaban et al. 2019), and we are somewhat unsure if it is widely understood. In any case, we agree that it is important to refer to previous literature that reported more than biphasic behavior. We note that it is rare that they are quantitatively characterized. For example, ref 16 did not provide a quantitative analysis of phases in the killing curve. In ref. 3, the wt kill curve was fitted with more than 2 exponentials, but one of the persister subpopulations was interpreted as the type-I persister, i.e., persistence induced in the stationary phase.

In the revised manuscript, we have updated the abstract and introduction to be clear that more than two phases have been discussed previously and also to include these references as examples of more than biphasic persistence.

Another comment refers to the triggered persistence assay: once the bacteria are exposed to starvation, even for a short duration, they cannot be considered as "log phase bacteria", so that these observations should be placed in the context of triggered persistence.

In the modified manuscript, we explicitly state that the starvation pulse induced persisters are triggered persisters.

Additional comments:

Effect of ppGpp mutants in reducing the triggered persistence of wt E.coli has been shown in Koch et al. *Molecular Microbiology* 2003.

We believe the reference meant is: S. B. Korch, T. A. Henderson and T. M. Hill, *Molecular Microbiology* vol. 50, p. 1199 (2003). We have included the reference as another example of relA dependent triggered persistence study.

Line 283 : killing curves up to 20 days in antibiotics have been done in the past (for example for MTB) so maybe avoid calling it a "new long-term persister assay".

We have removed “new” from the expression.

Please define how the different biological replicates are done. Are they started each from a separate single colony?

Yes, they started each from a separate single colony. We have updated the method section to explicitly state this.

Fig. S1: please define w/DS in legend

We have added the definition to the figure legend.

MOPS: is there any addition of trace elements or B1?

The trace metals were added but not B1. The E. coli strain used does not require a B1 supplement. We have added the composition of MOPS minimal medium to the supplement.

Reviewer #2 (Comments to the Authors (Required)):

GENERAL COMMENT:

In their paper, Svenningsen et al. explore the existence of long-term persisters in E. coli. The authors performed killing assays on a long time scale (1 week). Then, they assessed the role of RelA and the impact of nutrient starvation in the killing dynamic they observed.

Even if the first part of the manuscript about the long-term survival of bacteria has a potential interest, the general message needs to be clarify. The small amount of data and their relevance are questionable. The authors' conclusions could be entirely restricted to lab conditions. For example, RelA-dependent production of (p)ppGpp is required by pathogens to invade host cells and to establish a long-lasting infection (even in absence of antibiotics). Therefore, the role of RelA in this timescale is somehow questionable.

Our experiments are only performed in well-controlled laboratory conditions, and they do not immediately translate to the more complex situations such as the invasion process of the host. We updated the abstract so that it is clear that our experiments are in the laboratory condition.

The reviewer questions the role of RelA in persistence at a time-scale of a week, because one role of ppGpp is to establish a long-lasting infection. However, that does not necessarily contradict with the hypothesis that RelA also has a role in

persistence in the timescale of one week. It is open for future experiments to test if the RelA is also relevant for the persistence in the timescale of a week upon infection of a host, though designing such an experiment is not trivial.

In addition, the authors discussed about the log-phase persisters but they did not take in consideration the role of ciprofloxacin itself in persister formation.

The main focus of this study is the quantitative characterization of killing dynamics under well-controlled conditions. Hence, we have intentionally avoided the detailed discussion on the possible molecular mechanisms for persistence in the manuscript. However, responding to the comment, we decided to briefly mention the possibility and cite related literature (Dorr et al. Plos. Biol. 2010).

Finally, the last part about the role of RelA in this phenomenon is superficial and does not bring any insights on persister physiology.

Our work quantified the effect of RelA has on the persister kinetics over a week in the log-phase in different minimal media and after a carbon starvation pulse for the first time. Please note that our work was not the aim to clarify a molecular mechanism. We believe that it is a significant contribution to the field of persister physiology to quantify the persister kinetics and the effect of RelA in a well-controlled laboratory condition, especially because the field is far from concluding on molecular mechanisms of persistence despite so many attempts.

In conclusion, this piece of work is still preliminary. Therefore, we strongly recommend the authors to better investigate and strength the link between (p)ppGpp and long-term persistence by adding some controls (the relA/spoT mutant for example).

The main focus of this study is to quantify the long-term killing dynamics in a well-controlled log-phase culture and the effect of short starvation pulse on this. Please note that the difference between the triggered and spontaneous persistence was emphasized in Balaban et al. (2019) as a consensus of the field, and it pointed out that spontaneous persistence can only be quantified under well-controlled laboratory experiments. Our study also demonstrated the effect of RelA on this fundamental persister physiology. We have used minimal medium to investigate persister physiology in bacteria that are in the unsupplemented state of synthesizing their own building blocks (amino acids, nucleotides, etc). This condition prevents us from studying the relA/spoT double mutant under the same conditions, because this mutant cannot grow in unsupplemented minimal medium (see Potrykus et al. Environmental Microbiology (2011) 13, 563-575). Our study clearly showed in what situation relA has an effect on persistence. We have also quantified the (p)ppGpp level in relA mutant and wildtype. We believe that it is justified to discuss the link to (p)ppGpp given the ample evidence in the literature of the link between (p)ppGpp

and persistence, but this work is not meant to show evidence for the molecular mechanisms underlying this link.

ABSTRACT

- "...bacterial population is killed within a time scale comparable to their generation time when treated with a lethal concentration of antibiotics". This sentence is very confusing. What does "comparable to their generation time" mean?

We meant that the time scale of killing is similar to the bacterial generation time before the addition of antibiotics. We modified the expression.

INTRODUCTION

-Line37 - "One carefully executed study showed that no *E. coli* persister cells were formed during fast exponential growth in rich medium (7)". Please avoid using too many superlatives when referring to a study written by the same authors (M.A. Sorensen in both paper).

This reference is: I. Keren, N. Kaldalu, A. Spoering, Y. Wang, K. Lewis, Persister cells and tolerance to antimicrobials, *FEMS Microbiology Letters* 230 (2004) 13–18. doi:10.1016/S0378-1097(03)00856-5. The full author list is included and M. A. Sorensen is not an author.

-Title of the reference 9 is incorrect.

We have corrected it.

-Line64 - "low level contamination of the persister assay". Please clarify what "low level contamination" means.

We have rewritten the paragraph.

-Line78 - Reference 12 is not related to persisters. This paper refers to "non-heritable antibiotic resistance" (cfr. title) and do not fit with the standard definitions of the field (ref. 4). Please remove.

Balaban et al. (2019) defines antibiotic persistence based on the killing curve, where the subpopulation is killed at a significantly slower rate than the majority, and also non-heritable (i.e., not all the progeny of the antibiotic survivors will show the same level of tolerance when exposed to the same antibiotics again). This reference (Pontes and Groisman) presents the killing curve that satisfies the condition. Note that they call their phenomenon persistence, already clear in the abstract. The article was accompanied by a focus article entitled "Slow growth causes bacterial

persistence" by N. Kaldalu and T. Tenson (Science Signaling 12, Issue 592, eaay1167, DOI: 10.1126/scisignal.aay1167), which shows that other members of the field consider the phenomenon persistence. Therefore, we have kept this reference in the manuscript.

-Line90- remove "stress".

We have removed it. .

RESULTS

Long-term persister assay of exponentially growing cells

-Line128 - "OD436" - Why not OD600?

Because we are using Eppendorf photometers with fixed band filters, 600 nm is not an option for us. It is not uncommon in bacterial growth physiology literature to use wavelengths around 420-460 nm. See e. g. (Bremer and Dennis, (1987) In "*Escherichia coli* and *Salmonella typhimurium* Cellular and molecular biology", F.C. Neidhardt et al. eds . Washington D.C.: American Society for Microbiology, pp. 1527-1542; Potrykus et al., (2011) *Environ Microbiol* 13, 563-575.)

-Line133- The authors claim that *E. coli* persisters were formed during exponential phase. It would be nice if the authors could describe the method in a cartoon. 20 doubling times should be around 1000 minutes, which correspond to approximately 16h. Please precise this in the text. In addition, ciprofloxacin has been shown to induce bacterial persistence in *E. coli* (Dorr et al., 2010). Did authors try others AB (beta-lactams)?

In order to ensure the culture is in exponential growth, we made sure that the cells had experienced at least 20 doublings with keeping OD below 0.3 by series of back-dilutions. However, the actual time of the growth varies from culture to culture, so we do not give the actual time. Responding to the referee, we updated Figure 3A to include the growth step.

We performed several pilot experiments using beta-lactams, ampicillin and carbenicillin, but it turned out they are unstable under our condition, being degraded after 1-2 days. We have chosen ciprofloxacin because it stayed effective over a week-long experiment.

Line148-153- "The test rejects a double exponential as a good fit, but accepts the hypothesis of a triple exponential." Could you rephrase and clarify?

We have rephrased the text.

Line159- "the dominant phase of killing". What does "dominant" mean here?
Please, rephrase.

We have rephrased the text.

Line169- I would recommend to add - at least - some growth curves in minimal medium glucose or glycerol (for the WT and the recA mutant) to enrich the paper with some addition data.

We have added the growth curves in the supplement.

Line178-"the survival curve in glycerol medium at 1 to 2 days had a steeper drop than in glucose medium". This is not what I see on the figure 2A. The major difference between glucose and glycerol is between the day 0 and the day1 - under the consideration it is indeed significant. Could the authors clarify this point and perform statistical analysis to quantify the significance of the differences they observe?

Thank you for pointing this out. Our description was imprecise and indeed the significant drop is from 7 hours to 21 hours. We have rephrased the text.

Line183-"In fact, in two of the three biological replicates of WT cultures growing in glycerol, almost no survivors were observed after 3 days of killing". The discrepancy the authors describes by comparing their biological repeats is confusing. Please rephrase. In addition, by looking at figure 2A, no difference between glucose and glycerol at 3 days can be observed, which is not the conclusion of the authors. Please, clarify.

This sentence is part of the paragraph that explains that no significant difference was observed in the survival of WT in the glucose minimal medium and glycerol minimal medium after 3 days. It is to emphasize that survivors are not higher in the glycerol medium than glucose after 3 days. There is no discrepancy. In the revised manuscript, we explicitly emphasized this point.

We found that the last sentence of the paragraph that summarizes the fit was somewhat disconnected from the flow of the content, and it may have caused some confusion. Therefore, we restructured the paragraph, now it is made into 3 separate paragraphs.

Deletion of relA affects the killing dynamics

Line205- "However, in low energy, SpoT produces insufficient (p)ppGpp to suppress the growth rate when RelA is missing". What does "suppress the growth rate" mean?

We meant to reduce the growth rate. The text was updated.

Line212- the authors should add the double *relA*/*spoT* mutant to discuss the role of (p)ppGpp (and quantify (p)ppGpp in *relA* or *relA*/*spoT* mutant in their condition). We have used the minimal medium supplemented only by carbon source to have a well-controlled log-phase culture. This prevents us from studying the *relA*/*spoT* mutant in the same condition, because such mutant cannot grow in the media we used.

Line216- "(Fig. 1B inset)". Do you mean Fig. 2B inset?
Thank you for noticing this. We have corrected it.

Figure 2C - "The *relA* mutant had more survivors, but there was no statistically significant difference at any of the time point". If it is no statistically significant, the sole conclusion should be that *relA* mutant does not have more survivors. Please rephrase or clarify your point.

We agree that it was a confusing text. We rephrased the sentence to simply state that there was no statistically significant differences.

Figure2E - Two hours treatment does not allow any convincing conclusions regarding persistence, therefore, what is the message of this panel? If any, please remove it.

This data is meant to show that the initial rate of killing is correlated with the doubling time before addition of the antibiotics. Because the initial rate of killing affects when the second phase (persister killing rate) becomes visible, we think it is an important quantity to mention. We have added a sentence to explicitly summarize this.

A starvation pulse prior to the antibiotic application affects the long-term persistence of wt cells in glucose minimal medium.

Title: A starvation pulse done in a glucose minimal medium seems contradictory. Please rephrase.

We have modified the section title.

Line248 - It is unclear why the authors filtered their bacteria instead of washing 2 times with a minimal medium containing no carbon sources. Please explain.

We did it this way to make as little as possible interference with the cells' physiological state. By filtering the cells, the time spent without the growth medium is very short, on the order of 1-2 minutes. However, with washing, the cells will

experience much longer time without medium. There is also the increased risk of removing chunks of the population when removing the supernatant. In addition, the centripetal force might interfere with the physiology of cells and could potentially be a stressor.

Figure 3B - To assess the validity of their experiment, the authors should quantify (p)ppGpp in a *relA/spoT* mutant (as a control).

As explained above, the *relA/spoT* mutant cannot grow under the growth conditions used in this paper. However, to qualify the validity of the method and our ability to carry it out, we provide references to previous (p)ppGpp measurements carried out by us or members of our research group using the same method (Tian, C., et al. (2016). *Journal of bacteriology*, 198(14), 1918-1926; Sinha AK and Winther KS. (2021) *Communications biology* ;4(1):1-0) as well as measurements of ppGpp using this method from the Cashel laboratory, where ppGpp was discovered and first characterized: Sarubbi, E. et al. *Journal of Biological Chemistry*, 264(25), 15074-15082. Thin layer chromatography is thus the standard method of making ppGpp measurements, and a method that we are well acquainted with in the laboratory. We have added the phosphorimager scan of TLC plates in Supplementary Figure S7 as the original data for the (p)ppGpp measurement.

DISCUSSION

Line 286 - "Spontaneous persisters were observed". I would recommend the authors to discuss the effect of ciprofloxacin on persister formation.

We are aware of the study that showed ciprofloxacin induces persistence by activating toxin *tisB* through SOS response (Dorr, Vulic, Lewis, Plos Biol. 2010). Even though the detailed molecular mechanisms of the persistence is not a focus of the study, the information may benefit the readers who are interested in the molecular mechanisms. Therefore, we added a paragraph in the discussion section.

October 18, 2021

Re: Life Science Alliance manuscript #LSA-2021-01076-TR

Prof. Namiko Mitarai
University of Copenhagen
The Niels Bohr Institute
Blegdamsvej 17
København, København 2200
Denmark

Dear Dr. Mitarai,

Thank you for submitting your revised manuscript entitled "Existence of log-phase *Escherichia coli* persists and lasting memory of a starvation pulse" to Life Science Alliance. The manuscript has been seen by the original reviewers whose comments are appended below. While the reviewers continue to be overall positive about the work in terms of its suitability for Life Science Alliance, some important issues remain.

Our general policy is that papers are considered through only one revision cycle; however, given that the suggested changes are relatively minor, we are open to one additional short round of revision. Please note that I will expect to make a final decision without additional reviewer input upon resubmission.

Please submit the final revision within one month, along with a letter that includes a point by point response to the remaining reviewer comments.

To upload the revised version of your manuscript, please log in to your account: <https://lsa.msubmit.net/cgi-bin/main.plex>
You will be guided to complete the submission of your revised manuscript and to fill in all necessary information.

B. MANUSCRIPT ORGANIZATION AND FORMATTING:

Sincerely,

Reviewer #1 (Comments to the Authors (Required)):

The authors have addressed my comments.

Reviewer #2 (Comments to the Authors (Required)):

General comment:

In the revised version, the authors did not add any interesting data to improve their manuscript. Without taking into account numerous suggestions, the authors resubmitted the same manuscript with solely minor modifications in the text. The amount of data and the general message remain preliminary and light and required further work before publication in Life Alliance. The authors should therefore consider expanding their study by adding others antibiotics or other mutants for example.

TITLE

In their paper, the authors did not prove the existence of log-phase E. coli persisters but only suggested it. If the authors want to claim this, they should compare the killing curves of population at different dilution cycle (post stationary phase) as it was done in Harms et al., 2017.

ABSTRACT

INTRODUCTION

RESULT

Line 133 - The authors did not validate their conclusion with another antibiotic. They justified their decision by the instability of beta-lactams. If the antibiotic is unstable, did the authors observe growth at some point? Why not adding beta-lactams every 24h?

Line 169 - The figure S4 is unclear. First, the colour code is not described in the legend and this should be fixed. Second, the time frame of the experiment is short (around 2 hours). Usually, growth curves are calculated by measuring the OD every 15-30 minutes during at least 8 hours (to reach the stationary phase). Finally, the scale of the Y axis is unclear (OD? CFU?)

Dear Dr. Sawey,

Thank you for the review. We believe that we have addressed all the concerns of the reviewer in the reply listed below, and the manuscript is ready for publication. We look forward to hearing from you.

Sincerely yours,

Mikkel S. Svenningsen, Sine L. Svenningsen, Michael Sørensen and Namiko Mitarai

Reviewer #2:

General comment:

In the revised version, the authors did not add any interesting data to improve their manuscript. Without taking into account numerous suggestions, the authors resubmitted the same manuscript with solely minor modifications in the text. The amount of data and the general message remain preliminary and light and required further work before publication in Life Alliance. The authors should therefore consider expanding their study by adding others antibiotics or other mutants for example.

We believe that our work contributes significantly in understanding the long-term persistence in well-controlled laboratory conditions, as referee 1 also agreed in his/her first referee report. Of course, further work is necessary for the progress of the field and we believe that our work can motivate many new works on persistence.

TITLE

In their paper, the authors did not prove the existence of log-phase E. coli persisters but only suggested it. If the authors want to claim this, they should compare the killing curves of population at different dilution cycle (post stationary phase) as it was done in Harms et al., 2017.

Balaban et al. 2019 emphasises that (i) enough serial dilution to ensure that the effect of the carry-over cells is minimized, and (ii) ensuring that the bacterial density is kept low enough to prevent reentering the stationary phase are the two key points for measuring the spontaneous / log-phase persisters.

As described in the method section, in our experiments, the stationary phase culture is diluted at least by 10^9 fold via serial dilutions. The number of persisters reported in our experiments are well above the number of possible carry-over cells from the stationary phase. In addition, OD_{436} (optical density, measure of the bacterial growth) was kept below 0.3 by serial dilutions, well below the saturation (about 10-fold below). Therefore, the reported persisters must have originated from the growing log-phase E. coli culture.

For clarification, we have added these points to the manuscript.

ABSTRACT

INTRODUCTION

RESULT

Line 133 - The authors did not validate their conclusion with another antibiotic. They justified their decision by the instability of beta-lactams. If the antibiotic is unstable, did the authors observe growth at some point? Why not adding beta-lactams every 24h?

While we agree that it is very interesting to have the killing assay for various antibiotics, it is not a requirement for a persister assay to test multidrug persistence. However, we did actually aspire to include both ciprofloxacin and ampicillin in the original setup. We found that, in the ampicillin culture, the cells in the cultures often started to increase in numbers after a few days. We have also tried exactly what the reviewer suggests, to add ampicillin every day. However, the ampicillin effect was still unstable. In addition, we have been trying very hard to keep the growth conditions well-defined and constant as much as possible and adding ampicillin every day leads to a very varying ampicillin concentration. We further tried the supposedly more stable beta-lactam carbenicillin, but this was also unstable. We suspect that beta-lactamase was released into the cultures from all cells initially lysed. It should be noted that we did conduct one experiment, where the cellular number decreased for a full week, though this was more the exception than the rule. We did not pursue further to clarify why ampicillin was ineffective in a long term. All in all, ampicillin and beta-lactams did not seem suited for long-term persister experiments, at least with our strains and growth conditions.

Line 169 - The figure S4 is unclear. First, the colour code is not described in the legend and this should be fixed. Second, the time frame of the experiment is short (around 2 hours). Usually, growth curves are calculated by measuring the OD every 15-30 minutes during at least 8 hours (to reach the stationary phase). Finally, the scale of the Y axis is unclear (OD? CFU?)

The colour code for Fig. S4 is now explained explicitly. Each color represents a biological replicate. What is shown in Y-axis, OD_{436} , is the optical density of the culture measured at the wavelength of 436nm, which reflects the bacterial growth similar to the commonly used OD_{600} . We extended the figure legend to explain this.

In order to keep the cells in the exponential phase, we had to make several serial dilutions to keep the density of bacteria sufficiently low, because otherwise, some cells may enter the early stationary phase. More specifically, we made sure OD_{436} is kept below 0.3. Because of this, it was not possible to follow growth curves to reach the stationary phase as the referee suggested.

October 28, 2021

RE: Life Science Alliance Manuscript #LSA-2021-01076-TRR

Prof. Namiko Mitarai
University of Copenhagen
The Niels Bohr Institute
Blegdamsvej 17
København, København 2200
Denmark

Dear Dr. Mitarai,

Thank you for submitting your revised manuscript entitled "Existence of log-phase *Escherichia coli* persists and lasting memory of a starvation pulse". We would be happy to publish your paper in Life Science Alliance pending final revisions necessary to meet our formatting guidelines.

- please upload your supplementary figures as single files also
- please add an Author Contributions section to your main manuscript text
- please add a conflict of interest statement to your main manuscript text
- please add your main, supplementary figure, and table legends to the main manuscript text after the references section
- please consult our manuscript preparation guidelines <https://www.life-science-alliance.org/manuscript-prep> and make sure your manuscript sections are in the correct order and labeled correctly
- please use the [10 author names, et al.] format in your references (i.e. limit the author names to the first 10)
- please upload your Tables in editable .doc or excel format
- The figure legend for figure 2 is missing mention of Panel E
- please add callouts for Figures 4A, E and S1, S2, S3, S5, S7 to your main manuscript text
- the experiment and data analysis details from the Supplementary material should be incorporated into the main ms text. Our Materials and Methods section has no limit. The References from the Supplemental material should also be incorporated into the main Reference section. The figures and tables provided in the Supplement appear to be located appropriately.

A. FINAL FILES:

B. MANUSCRIPT ORGANIZATION AND FORMATTING:

Sincerely,

November 4, 2021

RE: Life Science Alliance Manuscript #LSA-2021-01076-TRRR

Prof. Namiko Mitarai
University of Copenhagen
The Niels Bohr Institute
Blegdamsvej 17
København, København 2200
Denmark

Dear Dr. Mitarai,

Thank you for submitting your Research Article entitled "Existence of log-phase *Escherichia coli* persists and lasting memory of a starvation pulse". It is a pleasure to let you know that your manuscript is now accepted for publication in Life Science Alliance. Congratulations on this interesting work.

DISTRIBUTION OF MATERIALS:

Again, congratulations on a very nice paper. I hope you found the review process to be constructive and are pleased with how the manuscript was handled editorially. We look forward to future exciting submissions from your lab.

Sincerely,
